# The Effects of Broadband Infrastructure on Carbon Emission Efficiency of Resource-Based Cities in China: A Quasi-Natural Experiment from the “Broadband China” Pilot Policy

**DOI:** 10.3390/ijerph19116734

**Published:** 2022-05-31

**Authors:** Bo Li, Jing Liu, Qian Liu, Muhammad Mohiuddin

**Affiliations:** 1School of Management, Tianjin University of Technology, Tianjin 300384, China; lb2088@email.tjut.edu.cn (B.L.); 15565543721lj@stud.tjut.edu.cn (J.L.); 2School of Public Health, Tianjin Medical University, Tianjin 300070, China; 3Faculty of Business Administration, Laval University, Quebec, QC G1V 0A6, Canada

**Keywords:** Broadband China, carbon emission efficiency, resource-based cities, DID

## Abstract

Resource-based cities (RBCs) face serious environmental pollution, and there are efforts to try to overcome those challenges by transforming industrial structure through investing in new technologies. Based on the panel data of 114 prefecture-level resource-based cities in China, this paper uses the difference-in-differences (DID) method to identify the effects of the “Broadband China” pilot policy on the carbon emission efficiency of resource-based cities. The results show that the “Broadband China” pilot policy has a significant effect on the improvement of carbon emission efficiency of resource-based cities, and the results are still valid after parallel trend test, PSM-DID estimation and placebo test. This study also finds that there are differences in the carbon emission efficiency of different locations and types of resource-based cities. In addition, the results of the mechanism analysis show that the “Broadband China” pilot policy can promote the improvement of carbon emission efficiency by promoting the upgrading of the industrial structure, the accumulation of human capital and the improvement of the level of urban innovation of resource-based cities. The findings provide a reference for China’s resource-based cities to develop the Broadband infrastructure, realize industrial upgrading, accumulate human capital and improve urban innovation level, and promote low-carbon transformation and improve carbon emission efficiency.

## 1. Introduction

With the rapid development of urbanization and industrialization, the control of carbon emissions has long been an important issue faced by all mankind [1,2]. China’s rapid industrial growth, and urbanization consume a lot of energy and resources. At the same time, share of heavy industry is very high in the national economy, and the factor supporting the huge heavy industrial production is the combustion of coal, which makes China gradually become the largest carbon emission country in the world. Facing the increasingly severe global climate change problem, China proposed the goal of “Carbon Peaking and Carbon Neutrality” in 2021 (referred to as the “dual carbon” goal), and implementing carbon emission reduction and improving carbon emission efficiency has become an important strategy to achieve ecological civilization building and green and low-carbon development. Advanced technological infrastructure such as Broadband and other digital network infrastructure can make the current activities more environment friendly as well as can help to transform the local economic structure. Cities are the core development hubs to achieve the “dual carbon” goal, and they are also the main fronts to deal with the above challenges. Resource-based cities, as important strategic bases for energy resources in China, are characterized by high carbon emissions and mostly sluggish economic development, and the proposed carbon neutrality target also brings new challenges and opportunities for the transformation of resource-based cities.

Specifically, resource-based cities, as special cities, have gradually developed into having an important economic status in China by relying on their own rich mineral resources development and operation. However, in recent years, as the global resource market has entered a recession, coupled with the gradual depletion of resources, the overall economic development of resource-based cities in China has been slow, and some cities have even experienced a decline. Among them, as a typical old industrial base and resource-based city agglomeration in China, the economic development situation in the three northeastern provinces is still unsatisfactory and the brain drain is serious, despite a series of special policies introduced by the state to revitalize the northeast. In addition, many resource-based cities still make articles around resource-based industries in the process of transformation, such as coal resource-based cities around coal vigorously develop coal power, coal chemical industry, coal machinery and equipment manufacturing industries, which has achieved certain economic benefits in the short term. However, in the long term, the transformation mode of extending the traditional industrial chain and relying on high energy consumption industries such as mineral resources mining and related energy industries to develop the economy will still lead to high carbon emissions in resource-based cities, and the transformation of resource-based cities to develop the economy in the context of carbon neutrality also faces the pressure of controlling carbon emissions. Therefore, a reasonable assessment of the carbon emission efficiency of resource-based cities is conducive to their low-carbon economic transformation and the achievement of China’s “dual carbon” goal.

The rise of the Internet has increased opportunities to transform economic development and promote sustainable development, and to a certain extent, it can alleviate environmental pressure [3]. As a key link in the construction of information infrastructure, broadband supports the development of the Internet of Things, cloud computing and other high-tech industries, plays an important role in the high-quality development of the economy and society, and has important strategic significance in the new round of technological revolution and industrial transformation integrating cyber physical systems (CPS) [4]. As an emerging market has tremendous challenge to develop private sector led digital infrastructure, Chinese government introduced “broadband policy” to push for developing rapid infrastructure. Broadband infrastructure refers to a series of facilities required in the process of obtaining broadband network or broadband network construction, which includes broadband network infrastructure involved in the adoption of wired access such as xDSL and FTTx and wireless access such as 3G, 4G, LTE and broadband satellite, etc., and other mobile communication technology and wireless access related infrastructure. The development of broadband networks allows the sharing of effective information in environmental protection at any time, improving the efficiency of environmental monitoring and information management, and realizing the organic combination of environmental protection and information technology. And the improvement of broadband facilities provides hardware foundation for the development of information technology, which can drive the rapid development of new business models such as digital economy and information technology applications, while eliminating backward industries with high energy consumption and serious pollution, and improving the overall efficiency of key industries and ecosystems. At the same time, with the vigorous development of information technology, the improvement of broadband infrastructure can also help break the time and space limitations of environmental governance, realize dynamic environmental monitoring, real-time risk assessment and timely feedback [5], and improve the overall environmental governance efficiency. Currently, an increasing number of countries and regions have launched Internet strategies. Among them, China explicitly proposed the concept of new infrastructure construction (hereafter referred to as new infrastructure) for the first time in 2018, stating that it should “accelerate the pace of 5G commercialization and strengthen the construction of new infrastructure such as artificial intelligence, industrial Internet, and Internet of Things”. Resource-based cities usually regard the mining of mineral resources as the pillar industry, along with the interference of mining activities, the green infrastructure network structure of resource-based cities is also damaged to different degrees, promoting the construction of broadband infrastructure and vigorously developing the Internet is the key path to solve the environmental problems of resource-based cities.

In this context, it is particularly important to objectively and accurately assess the effect of broadband infrastructure construction on economic and social development. The “Broadband China” strategy has elevated broadband facilities to national strategic public infrastructure for the first time, which helps regions to build and improve broadband infrastructure. Therefore, taking the “Broadband China” pilot policy as the entry point to study the construction of broadband infrastructure and its relationship with carbon emission efficiency of resource-based cities is of great theoretical and practical significance to explore new paths and new models for low-carbon economic transformation and high-quality development.

## 2. Literature Review

Regarding carbon emission efficiency, domestic and foreign scholars have conducted extensive research, mainly focusing on the analysis of carbon emission efficiency measurement. Tone [6,7], by constructing a measurement model based on slack variables and considering undesired outputs, for the first time added slack variables to the constructed objective function to effectively measure environmental efficiency. Fare [8] and Fukuyama et al. [9] proposed a more general SBM directional distance function based on Tone’s study. Ramanathan et al. [10,11,12] used the DEA method to measure the carbon emission efficiency of 17 countries in the Middle East and North Africa, and analyzed the relationship between GDP, energy consumption and carbon emissions. Herrala et al. [13] used the SFA model to measure the carbon emission efficiency of 170 countries from 1997 to 2007, and their findings indicated that China had the lowest carbon emission efficiency. Zhou et al. [14] used the Malmquist carbon index to study the emission efficiency of the 18 countries with the highest global carbon emissions and analyzed the factors influencing the MCPI. Domestic scholars also used similar methods to study carbon emission efficiency, among which data envelopment analysis, stochastic frontier analysis and distance function were commonly applied. Zha et al. [15] used DEA model to construct carbon emission performance indexes from static and dynamic aspects respectively, and measured the industrial carbon emission performance of 30 provinces in China. Li et al. [16] explored the characteristics and differences of energy carbon emission efficiency in different economic regions of China using three-stage DEA and three-stage DEA-Malmquist index. Zhang et al. [17]. Used a stochastic frontier analysis model to evaluate and analyze the carbon emission efficiency of 30 provinces and the Yangtze River Delta region in China, respectively. Li et al. [18] measured the carbon emission efficiency by using the scale direction distance function considering slack variables. From this, it can be seen that the measurement of carbon emission efficiency is mainly focused on inter-provincial level and industry level, but there is no microscopic study on city level, and the research method is mainly based on DEA model.

In recent years, resource-based cities have gradually become the object of carbon emission research in academia, and the studies on carbon emission in resource-based cities are mainly divided into two categories. One category is to study all resource-based cities as research objects, such as Feng et al. [19] based on Chinese high spatial resolution grid data (CHRED), combined with DPSIR model, classification comparison and scenario analysis methods, systematically analyzed the CO_2_ emission characteristics of 126 resource-based cities in China, and analyzed the CO_2_ emission trends and emission reduction potential of resource-based cities in the future. Sun et al. [20] selected 106 resource-based cities in China according to their classification, and studied their carbon emission efficiency by combining the DEA model and SE-SBM model. Another type of study is to investigate the similarities and differences of carbon emission characteristics and drivers between modern industrial cities and traditional resource-based cities, such as Baotou and Wuxi, respectively. Another type of research is conducted with each resource-based city as the object of study. Zhang et al. [21] take modern industrial cities and traditional resource-based cities as the research objects, and select Baotou and Wuxi as the representative cities to investigate the similarities and differences of carbon emission characteristics and drivers of the two types of cities respectively. Chen and Sun [22] used eight petroleum-based resource cities as research objects to calculate the ecological deficit of petroleum-based resource cities based on carbon emissions from energy consumption and carbon carrying capacity of different land uses, and measured the spatial carbon footprints of different industries by combining different industrial spatial divisions. In general, although the research related to carbon emissions of resource-based cities is gradually abundant, the research area for resource-based cities at the prefecture level and above still needs to be expanded.

The role of information and communication technologies (ICT), especially broadband, in the development of individuals and socio-economics has been widely discussed [23,24,25]. Several studies have argued that ICT penetration has driven the development of the digital economy, which in turn has driven economic and technological paradigm change. Thus, broadband infrastructure or ICT penetration has grown by leaps and bounds and has a non-linear impact on economic, social and environmental systems [26,27]. Broadband Internet access and broadband infrastructure improvements have profoundly changed all aspects of China’s economic development, and unlike traditional infrastructure such as roads and bridges, which are significantly different, broadband can fundamentally affect the way economic activities are organized and run, Koutroumpis [28] found increasing returns on broadband investments based on data from 22 OECD countries from 2002 to 2007, and Czernich et al. [29] point out that broadband can facilitate the development of new business models by facilitating the spatial dissemination of information, which in turn reduces industry entry costs and increases market transparency. In terms of environmental protection, most scholars focus on the impact of traditional infrastructure represented by transportation infrastructure on environmental pollution, Yang et al. [30] showed that the opening of high-speed rail suppressed sulfur dioxide emissions from urban industries in China, Dalkic et al. [31] took high-speed rail in Turkey as an example, and found that the opening of high-speed rail reduced carbon emissions, and some studies concluded that urban rail transit, electric vehicles, and rapid transit systems (BRTs) were more effective in reducing carbon emissions. Some studies have also suggested that urban rail, electric vehicles, and bus rapid transit (BRT) systems can also help reduce air pollution [32,33].

At present, as the network information infrastructure and carbon emissions have gradually attracted the attention of various countries, the relationship between the two has also become a hot research topic in academia. According to the existing theoretical research [34,35,36], the development of the Internet can change the transaction space, prolong the transaction time, expand more transaction channels, save unnecessary intermediate links, and have a wide-ranging and far-reaching influence on activities such as production and operation, transaction distribution, and organizational behavior, thereby improving the energy utilization efficiency of enterprises, which indirectly improves the efficiency of carbon emissions. And with the increasing integration of the Internet and industrial development, the role of the Internet in improving energy efficiency has been confirmed by more and more empirical studies [37,38,39,40,41,42]. However, few scholars directly study the relationship between network information infrastructure and carbon emission efficiency, and most of them discuss it indirectly. Some scholars believe that the rapid development of Internet-based information and communication technology and related industries leads to the rapid growth of electricity consumption [43], thereby promoting the increase of carbon emissions [44]. Other scholars believe that the development of information and communication technology will improve environmental quality by reducing greenhouse gas emissions. For example, some studies have shown that the use of the Internet and the increase in Internet penetration will significantly reduce carbon emissions in the long run [45,46]. Increasing investment in information and communication technology infrastructure also has a significant effect on reducing carbon emissions [47], and the carbon trading platform based on digital information technology can also reduce the cost of carbon trading [48], and accelerate the reduction of carbon emissions and carbon emission intensity.

It can be seen that the existing research has laid an important foundation for understanding the relationship between information infrastructure construction and carbon emission efficiency. However, the analytical framework, research objects and research methods for studying the relationship between information infrastructure construction and carbon emission efficiency still have certain limitations and need to be further deepened. First, in terms of analytical framework, the existing research mainly focuses on the carbon emission reduction effect of Internet infrastructure development, and few scholars have conducted in-depth studies on its influence mechanism. Second, in terms of research objects, the existing research mainly studies carbon emissions from the macro perspective of countries, regions or industries, and rarely evaluates the influence of information/digital infrastructure on carbon emission efficiency from the perspective of cities (i.e., resource-based cities) with obvious high carbon emissions characteristics. Third, in terms of research methods, most of the existing research on the development of information infrastructure uses a single indicator for measurement, and the analysis process rarely involves the discussion of endogeneity. 

Therefore, based on the exogenous policy influence of “Broadband China”, this paper examines the effects of information infrastructure development on carbon emission efficiency and its influence mechanism from the city level, in an attempt to supplement the existing literature. The marginal contributions of this paper are mainly reflected in: First, this paper uses the DID method to evaluate the effects of “Broadband China” pilot policy on the carbon emission efficiency of resource-based cities from the city level. To a large extent, the endogeneity problems faced by research at the micro-level are avoided. At the same time, in order to further test whether there is an endogeneity problem in the regression results, this paper conducts a series of robustness tests to ensure the reliability of the empirical results. Second, this paper further examines the difference in the influence of “Broadband China” pilot policy on the carbon emission efficiency of different locations and different types of resource-based cities. Third, based on the mediation effect model, this paper analyzes the mechanism by which “Broadband China” pilot policy promotes the upgrading of industrial structure, the accumulation of human capital, and the improvement of urban innovation levels of resource-based cities, thereby promoting carbon emission efficiency.

## 3. Research Hypotheses

### 3.1. “Broadband China” Pilot Policy Background

With the vigorous development of the information industry, broadband, as a carrier of information, plays a fundamental role in economic development that cannot be ignored [49,50]. At present, more and more countries have formulated and implemented broadband policies. Among them, developed countries such as the United States, France and South Korea have successively introduced new broadband development strategies to seize the development opportunities of the digital economy [51,52,53]. Trends in broadband strategies have spread from developed to developing countries. International Telecommunication Union (ITU) called on all countries to have a viable broadband implementation plan by 2020 [54]. Since China’s first access to the Internet in 1994, China’s Internet infrastructure has achieved rapid development in many fields, which is partly due to the Chinese government’s high priority on broadband infrastructure development. In 2013, the State Council issued the “Notice of the State Council on the Issuance of the ‘Broadband China’ Strategy and Implementation Plan” to strengthen strategic guidance and systematic deployment to promote the rapid and healthy development of broadband infrastructure in China. 

The “Broadband China” strategy puts forward phased development goals, which include: first, to achieve full speed-up development by the end of 2013. Focus on strengthening the construction of fiber optic network and 3G network, increasing the access rate of broadband network, improving and enhancing the user’s Internet experience; second, from 2014 to 2015, vigorously promote the popularization. By 2015, the basic realization of urban fiber to the building into the home, rural broadband into the countryside into the village, fixed broadband household penetration rate of 50%, 3G and its long-term evolution technology (3G/LTE) user penetration rate of 32.5%, the proportion of administrative villages with broadband reached 95%, schools, libraries, hospitals and other public welfare institutions to basically achieve broadband access. At the same time, we will strengthen the research and development of new technologies in key areas such as optical communications, broadband wireless communications, next-generation Internet, next-generation radio and TV networks, and cloud computing, and achieve original innovation results in some key areas. Third, from 2016 to 2020, to achieve optimization and upgrading. Focus on promoting broadband network optimization and technology evolution and upgrading, broadband network service quality, application level and broadband industry support capacity to reach the world advanced level. In addition, the “Broadband China” pilot policy as an important “Broadband China” strategic special action to implement the “Broadband China” strategy as the core, the construction of “Broadband China” demonstration cities as a precursor to the implementation of the two mandatory national standards of fiber to the home as a grasp to promote the cities and counties in urban areas with full coverage of fiber optic network, and constantly expand the number of Internet broadband access ports, 4G base stations, fiber transformation of old neighborhoods, in order to solidly promote the construction of broadband cities. To this end, in 2014, 2015 and 2016, around 119 “Broadband China” demonstration cities were selected in three batches after comprehensive consideration of regional distribution, geographical conditions, and existing network facilities (Figure 1). After being selected as a “Broadband China” pilot city (group), the city will further focus on improving broadband network speed, expanding network coverage and expanding the scale of broadband users. It is foreseeable that the “Broadband China” pilot policy will directly promote the development of Internet in the city and facilitate the dissemination of local information and knowledge. At the same time, the selection of the “Broadband China” pilot cities is not influenced by the local ecological conditions, which avoids the problem of reverse causality in the policy and provides a basis for this paper to analyze the political impact of broadband infrastructure construction.

### 3.2. The Direct Effect of the “Broadband China” Pilot Policy on Carbon Emission Efficiency of Resource-Based Cities in China

At present, informatization and digitalization have gradually become a breakthrough topic in China’s economic and environmental governance [53]. Broadband networks and services are playing an increasingly important role in various fields of economy, society and life. On the one hand, knowledge spillover, technology spillover and economic agglomeration from informatization and digitization promote the technological progress [38,55]. As the foundation of optimizing the network development environment, the construction of broadband infrastructure is conducive to improving the overall network information technology service level, promoting the development of digital economy and further driving technological innovation. Among them, the Internet of Things, which is constantly developing based on broadband facilities, is a new information technology means, which can reduce energy consumption, save resources and improve efficiency through the perception and intelligent management of the physical world. In addition, the continuous improvement of broadband infrastructure lays a hardware foundation for digital transformation, which can drive the improvement of intelligent manufacturing infrastructure and promote green and low-carbon development. For example, intelligent manufacturing workshop can realize carbon emission prediction and low-carbon control based on digital data fusion drive [56]. On the other hand, “Broadband China” has accelerated the development of China’s telecommunications infrastructure, brought new vitality to economic development under the influence of the epidemic, and may play a role in the green development of all industries [29,30,31,32,33,34,35,36,37,38,39,40,41,42,43,44,45,46,47,48,49,50,51,52,53,54,55,56,57]. At the same time, the network infrastructure brought by “broadband China” plays an important role in accelerating the flow of innovation elements, reducing the spatial barriers of information transmission [58], promoting the application of innovative technologies [59], and promoting the realization of regional technological innovation development. Technological innovation can effectively realize the transformation of industrial development mode, especially for resource-based cities with extensive development mode, the construction and improvement of broadband infrastructure will effectively promote the optimization and upgrading of industrial structure, and existing studies show that the upgrading of industrial structure can effectively reduce carbon emissions [60]. The industrial sector concentrates most of the energy-intensive sectors [61], so its energy mix is significantly different from that of the service sector; this means that the high-level gradually in the process of industrial structure, mainly fossil energy structure can be improved, help reduce carbon emissions city, raise efficiency of carbon emissions. Therefore, hypothesis 1 is proposed:

**Hypothesis** **1** **(H1).**
*“Broadband China” pilot policy plays a positive role in promoting carbon emission efficiency of resource-based cities in China.*


Resource-based cities have obvious stage characteristics in the development process. Different resource-based cities face different political backgrounds, geographical characteristics and humanistic thoughts [62], and their development characteristics are significantly different According to their own resource security capabilities and development problems, they can be divided into four stages: growth stage, mature stage, decline stage, and regeneration stage, and different types of resources in different development stages. Different types of cities have different natural resource stocks, and there are also certain differences in the degree of resource development and economic development. In addition, resource-based cities in different regions have different natural resource endowments, and there are also great differences in the economic development of eastern and central and western regions, as well as coastal and inland cities. Therefore, this paper assumes that the effects of “Broadband China” pilot policy on the carbon emission efficiency of resource-based cities may also be different. Therefore, hypothesis 2 is proposed:

**Hypothesis** **2** **(H2).**
*The influence of “Broadband China” pilot policy on carbon emission efficiency of different locations and types of resource-based cities is heterogeneous.*


### 3.3. The Indirect Effect of the “Broadband China” Pilot Policy on Carbon Emission Efficiency of Resource-Based Cities in China

(1) “Broadband China” pilot policy affects the carbon emission efficiency of resource-based cities by promoting the upgrading of industrial structure.

The promotion effect of the “Broadband China” pilot policy to accelerate industrial structure upgrading is mainly reflected in the development of new industries and transformation of traditional industries. On the one hand, the “Broadband China” pilot policy promotes the continuous improvement of Internet-based network infrastructure construction and the emergence of new economic growth modes, such as new industries, new business modes and new models, which will help the development of a new generation of information technology and give better play to the role of data as a factor of production, drive industrial upgrading and promote the optimization of industrial structure. For example, in the field of services, intelligent transformation of services will be realized to promote the formation of a new industrial production and service system, which is conducive to optimizing the industrial structure of resource-based cities and realizing the transformation of traditional industries with high energy consumption, high carbon emissions and high pollution to new green industries with low energy consumption, low carbon emissions and low pollution, thus solving the contradiction between industrial development and environmental protection [63] and promoting their low-carbon development. In addition, the development of network infrastructure also empowers and triggers the evolution of the sharing economy, which provides traditional industries with accurate supply and demand matching [64] and promotes the upgrading and transformation of traditional industrial structures. On the other hand, industrial structure upgrading can indirectly increase carbon intensity by optimizing the division of labor and promoting technological change [65], which in turn reduces carbon emissions to promote low-carbon economic development. The upgrading of industrial structure promotes the proportion of tertiary industry to rise, and the continuous upgrading and innovation of existing products and technologies help to reduce emissions and protect the environment, thus improving the carbon emission efficiency of cities.

(2) “Broadband China” pilot policy affects the carbon emission efficiency of resource-based cities by promoting the accumulation of human capital.

The effects of the implementation of the “Broadband China” pilot policy on the accumulation of human capital is mainly reflected in the rich information resources brought by the continuous improvement of information infrastructure. With the implementation of the “Broadband China” strategy, it has promoted the continuous improvement of Internet-based network infrastructure construction and accelerated the progress of urban information technology, which is conducive to the formation of various data resource platforms, better collection, storage, development and utilization of various resources and data, and has a positive effect on innovative organizations [66,67], and is conducive to create a demonstration base for scientific and technological innovation and a highland for the gathering of scientific and technological talents. At the same time, network infrastructure has the ability to disseminate information across time and space, and the Internet can accelerate the cross-regional flow of talent elements, which can integrate various advantageous resources on a wider scale, thus attracting more high-quality talent inflow and accumulating reserve forces for achieving green innovation development.

(3) “Broadband China” pilot policy affects the carbon emission efficiency of resource-based cities by promoting the improvement of innovation level.

The implementation of the “Broadband China” strategy can effectively improve the innovation level of resource-based cities. On the one hand, broadband infrastructure construction promotes the development of the Internet, which has the ability to accelerate the cross-regional integration of capital, talent, technology and other innovation factors. This will lead to knowledge spillover effects and encourage the proliferation of technological innovation systems characterized by manufacturing, learning, research and use. On the other hand, with the improvement of broadband infrastructure, enterprises’ broadband network capabilities and network-based process re-engineering and business innovation capabilities are gradually increasing, which drives technological advances in manufacturing, energy conservation, and environmental protection, and will help enterprises to enhance their technological innovation capabilities in research and development (R&D) and production sectors [68]. Technological innovation, in turn, promotes the advancement of manufacturing and energy-saving technologies, drives enterprises to replace traditional resource-intensive products with technology-intensive products, and fundamentally improves their resource utilization and manufacturing efficiency, significantly contributing to the green development of cities [69]. In addition, the implementation of broadband infrastructure construction policies represented by the construction of “Broadband China” demonstration cities is often accompanied by a large number of innovation subsidies and tax relief policies supported by the central and local governments, which can attract a large number of high-tech and digital innovation enterprises to gather and give birth to and accelerate the development of a new generation of information technology industry represented by cloud computing and big data. The development of information technology industry, represented by cloud computing and big data, can accelerate digital technology empowerment and promote urban innovation [70], thus achieving innovation-driven carbon emission efficiency increase.

Based on the above analysis, hypothesis 3 is proposed:

**Hypothesis** **3** **(H3).**
*“Broadband China” pilot policy can improve the carbon emission efficiency of resource-based cities in China by promoting the upgrading of industrial structure, accumulation of human capital, and the improvement of innovation level.*


## 4. Research Design

### 4.1. Data

This paper takes 116 prefecture-level resource-based cities among the 262 resource-based cities identified in the National Sustainable Development Plan for Resource-based Cities (2013–2020) issued by the State Council in 2013 as the research objects. In view of the completeness and availability of the data, this paper excludes Bijie and Jinchang from the urban sample, and uses the mean method to supplement the missing data, and finally selects the annual data of 114 resource-based cities from 2004 to 2018 as the research sample, with a total of 1710 observations. The data are mainly from the “China Statistical Yearbook”, “China City Statistical Yearbook”, “Urban Construction Statistical Yearbook” from 2005 to 2019, and statistical bulletins of national economic and social development of prefecture-level cities. Some missing data are filled using the mean method.

### 4.2. Model

This paper takes the construction of the “Broadband China” pilot policy as a quasi-natural experiment and uses the DID method to test the influence of the “Broadband China” pilot policy on the carbon emission efficiency of resource-based cities. Since the “Broadband China” demonstration cities are determined in batches, this paper draws on the research of Beck et al. [71] to construct the progressive model as follows:(1)$$CEE_{it}=\alpha_{0}+\alpha_{1}{\mathrm{broadband}}_{it}+\sum_{j}\beta_{j}Control_{it}+\gamma_{t}+\eta_{i}+\varepsilon_{it}$$
where the subscripts *i* and *t* represent city and time, respectively. *CEE_it_* is the core explained variable of this paper, representing the carbon emission efficiency of the *i*th city in the *t* year. *broadband**_it_* represents whether city *i* is determined as the “Broadband China” demonstration city in the *t* year. If a city is determined as the “Broadband China” demonstration city in the current year and subsequent years, let *broadband**_it_* = 1; if the city is not identified as a “Broadband China” demonstration city, then *broadband**_it_* = 0. *Control_it_* represents the other control variables affecting the carbon emission efficiency of resource-based cities with time and individual changes, *γ_t_* represents the time-fixed effect, *η_i_* represents the fixed effect of individual cities, and *ε_it_* is the random error term.

In order to further identify the intermediate mechanism of the “Broadband China” pilot policy affecting the carbon emission efficiency of resource-based cities, this paper draws on the practice of Baron and Kenny [72] and uses the intermediate effect model to test. Specifically, based on model (1), the following recursive model is constructed:(2)$$M_{it}=\varphi_{0}+\varphi_{1}{\mathrm{broadband}}_{it}+\sum_{j}\beta_{j}Control_{it}+\gamma_{t}+\eta_{i}+\varepsilon_{it}$$
(3)$$CEE_{it}=\lambda_{0}+\lambda_{1}{\mathrm{broadband}}_{it}+\lambda_{2}M_{it}+\sum_{j}\beta_{j}Control_{it}+\gamma_{t}+\eta_{i}+\varepsilon_{it}$$
where *Mid_it_* represents the intermediary variable, which is industrial structure upgrade (*industrs*), human capital (*humancs*) and innovation level (*innova*), respectively, and the meanings of other variables are the same as model (1). The mediation effect of the model test is divided into three steps: The first step is to estimate the coefficient *α*_1_ of the model (1), and test the total effect of the “Broadband China” demonstration cities construction on the carbon emission efficiency of resource-based cities. If *α*_1_ is significantly positive, it means that the “Broadband China” demonstration cities construction can significantly improve carbon emission efficiency of Resource-Based Cities. In the second step, the coefficients *φ*_1_ and *λ*_2_ of model (2) and model (3) are estimated, respectively. If both are significant, this indicates the existence of a mediating effect. On this basis, if *λ*_1_ is not significant, it indicates that the mediating variable has played a role of complete mediation; if *λ*_1_ is significantly positive and less than *α*_1_, it indicates that the mediating variable has played a partial mediating role. Finally, if at least one of the coefficients *φ*_1_ and *λ*_2_ is not significant, the Sobel test is used to judge whether the mediating effect exists or not.

### 4.3. Variable Selection

(**1**) **Explained variable:** carbon emission efficiency (*CEE_it_*). The current measurement of carbon emission efficiency is mainly through the construction of carbon emission input-output index system and the use of DEA and SFA models to measure the efficiency [10,11,12,13]. In the study of carbon emission efficiency measurement in resource-based cities, DEA model is chosen for the measurement in order to avoid the structural bias caused by the wrongly set production function. Meanwhile, considering the limitation of DEA model makes it impossible to further analyze the efficient decision units; while the super-efficiency model solves this problem well, and the SBM model appropriately deals with the non-expected output, so the analysis of super-efficiency SBM model is more accurate. In view of this, this paper measures the carbon emission efficiency of China’s resource-based cities based on the input-output perspective, and constructs a super-efficiency SBM model (See Appendix A for details) based on undesired output with reference to the relevant practices of Wang et al [73].

The fixed capital stock is selected as the capital input, referring to the practice of Zhang et al. [74] taking 2004 as the base period for perpetual inventory processing of fixed asset investment. The number of employees in each city at the end of the year is selected as a labor input. The whole urban social electricity consumption is selected as energy input. GDP was selected as the expected output in the process of economic activities, and the year 2004 was taken as the base period for the reduction treatment. The city CO_2_ emission is regarded as the undesirable output of economic activities, considering that the measurement content of CO_2_ emissions at the city level currently mainly includes liquefied petroleum gas, natural gas, electricity consumption and transportation carbon emissions, and the transportation carbon emission measurement data is seriously missing and its proportion in the total carbon emissions is small, so this paper comprehensively considers the consumption of liquefied petroleum gas, natural gas and electricity to calculate the total CO_2_ emissions of resource-based cities. And the specific calculation method refers to the calculation of CO_2_ emissions by Ren et al [75]. Table 1 shows the specific input–output index system of city carbon emission efficiency.

(**2**) **Core explanatory variable:** “Broadband China” pilot policy variable (*broadband*). This paper adopts the form of policy dummy variable to set. Specifically, in the year when a resource-based city is selected as the “Broadband China” demonstration city and the years after, the value of broadband is 1; otherwise, the value of broadband is 0. The Ministry of Industry and Information Technology (MIIT) and the National Development and Reform Commission (NDRC) designated 119 broadband China demonstration cities in 2014, 2015, and 2016, including 36 resource-based cities. Due to the serious lack of data in some cities, this paper excluded Bijie and Jinchang. Finally, 36 resource-based cities identified as “Broadband China” demonstration cities are included in the research sample, which constitutes the treatment group of the study sample, and the other 78 resource-based cities are the control group.

(**3**) **Mediating variables:** ① Industrial structure upgrading (*industrs*). Industrial structure upgrading, as a key way to achieve low-carbon economic development, reflects the process in which the focus of urban economic development and industrial structure gradually shifts from low level to high level. In this paper, the proportion of output value of the tertiary industry in GDP is selected to represent industrial structure upgrading. ② Human capital (*humancs*). Human capital is an important driving force of urban innovation activities; in particular, senior human capital is conducive to promoting the development of green and low-carbon technology innovation, thus reducing carbon emissions and improving carbon emission efficiency. This paper selects the proportion of students in institutions of higher learning to measure the stock of human capital in cities. ③ The level of innovation (*innova*). Urban technological innovation is the core driving force to realize low-carbon transformation and promote carbon emission efficiency. Therefore, the proportion of urban invention patents in the total patents is selected in this paper to represent the innovation level of a city.

(**4**) **Control variables.** This paper mainly controls the following variables in the empirical research process: ① Transportation (*traff*). As a basic and leading industry in the national economy, the transportation, storage and postal industry is also a major contributor to energy consumption and direct carbon emissions, and is closely related to urban economic development and environmental pollution, its scale plays an important role in the development of low-carbon urban transformation, and the number of employees in the transportation, storage and postal industry can, to a certain extent, reflect the size of the industry, so this paper selects the number of employees in the transportation, storage and postal industry as a representative indicator to measure the level of transportation. ② Employment level (*unemprate*). The unemployment rate of a city reflects the employment situation of the city, and the change of employment status will affect the change of household consumption embodied carbon emissions, and then affect the carbon emission efficiency. Therefore, this paper refers to existing studies [76] and chooses the urban registered unemployment rate to characterize the employment level of cities. ③ Highway Infrastructure (*perra*). There is a direct relationship between road infrastructure on traffic CO_2_ emissions and the increase of population density, and the increase of population density will reduce the level of CO_2_ emissions per capita, so this paper refers to the existing studies [77], and chooses the urban road area per capita to measure the road infrastructure situation in resource-based cities. ④ City scale (*cityscale*). The city scale reflects the level of urbanization development to a certain extent, among which the city population size is directly related to carbon emission, so this paper refers to the existing studies [78] and uses the year-end city municipal area population share to characterize. ⑤ Investment in education (*education*). To a certain extent, education investment reflects the importance that the city attaches to education and the degree of human capital demand, and human capital can accelerate the transformation of production factor allocation, improve economic efficiency, promote clean production, and thus have an important impact on carbon emission efficiency. Therefore, this paper selects the proportion of education expenditure to total fiscal expenditure to measure the education investment of the city. ⑥ Gross Domestic Product Growth rate (*iGDP*). The economic growth of resource-based cities is inextricably linked to the growth of mineral energy consumption and carbon emissions, so the GDP growth rate is chosen to characterize the economic development level of cities in this paper. ⑦ Physical capital level (*phycl*). Physical capital is an important guarantee to promote scientific and technological innovation and achieve economic transformation, which is conducive to the development and promotion of green and low-carbon technologies and will have a certain impact on carbon emission efficiency, so this paper uses the proportion of fixed asset investment to GDP to characterize the level of physical capital in resource-based cities with reference to the study of Liu et al. [79]. The descriptive statistics are presented in Table 2.

## 5. Empirical Analysis

### 5.1. Benchmark Analysis

The regression results of the influence of “Broadband China” demonstration city construction on the carbon emission efficiency of resource-based cities are shown in Table 3. Among them, the regression results without adding any control variables in column (1) show that the coefficient of “Broadband China” demonstration city construction is significantly positive, which preliminarily indicates that the construction of a “Broadband China” demonstration city will promote the improvement in carbon emission efficiency of resource-based cities. The regression results of adding control variables in column (2) show that the regression coefficient of broadband has not changed substantially, and it has passed the significance test at the 5% level, which once again shows that the construction of “Broadband China” pilot policy is conducive to improving the carbon emissions efficiency of resource-based cities. The above estimation results well verify Hypothesis 1, indicating that “Broadband China” pilot policy plays a positive role in promoting the carbon emission efficiency of resource-based cities in China.

### 5.2. Robustness Test

#### 5.2.1. Parallel Trend Test

Satisfying the assumption of parallel trend is a prerequisite for the use of the DID method. Therefore, to evaluate the implementation effect of “Broadband China” pilot policy, it is necessary to conduct parallel trend test on the explained variable, the carbon emission efficiency of resource-based cities, that is, to test whether there is a significant difference in the trend of carbon emission efficiency of resource-based cities before they are identified as “Broadband China” demonstration cities. Therefore, referring to existing research [80], this paper adopts the event study method to judge whether there is difference in trend change between the experimental group and the control group in the quasi-natural experiment of “Broadband China” demonstration city construction. If there is no significant difference, parallel trend test will be carried out, and the specific model is set as follows:(4)$$CEE_{it}=\alpha_{0}+\alpha_{1}{\mathrm{broadband}}_{it}^{-4}+\alpha_{2}{\mathrm{broadband}}_{it}^{-3}+\ldots +\alpha_{9}{\mathrm{broadband}}_{it}^{4}+\sum_{j}\beta_{j}Control_{it}+\gamma_{t}+\eta_{i}+\varepsilon_{it}$$
where, *broadband_it_*^±*j*^ is a series of policy dummy variables, and its specific value rules are as follows: When resource-based cities in the treatment group are identified as “Broadband China” demonstration cities *j* before (after), the value of *broadband_it_*^−*j*^ (*broadband_it_^j^*) is 1; otherwise, it is 0. The meanings of other variables are the same as those of model (1). In particular, *broadband_it_*^4^ denotes the treatment group that was identified as a “broadband China” demonstration city for 4 years or more. Table 4 and Figure 2 show the regression results of model (4) in different ways. It can be seen that before the implementation of the “Broadband China” pilot policy (i.e., *j* < 0), the estimated coefficients of all policy dummy variables are not significant. This shows that, before the implementation of the policy, there is no systematic difference in the trend of carbon emission efficiency of resource-based cities in the treatment group and the control group, and the parallel trend assumption is satisfied.

#### 5.2.2. PSM-DID

In order to further control the influence of the difference between the demonstration cities identified as “Broadband China” and the resource-based cities not identified as “Broadband China” demonstration cities under the “Broadband China” pilot policy on carbon emission efficiency, referring to Rosenbaum and Rubin [81], this paper is based on the propensity score matching method (PSM) reconstructs the control group to reduce the problem of sample selection bias. The specific operation steps are as follows: First, through Logit regression between the dummy variables and control variables between groups, estimate the probability of a city being identified as a “Broadband China” demonstration city, and obtain the propensity of each resource-based city to be identified as a “Broadband China” demonstration city. Second, the kernel matching method is used to find cities with the same propensity score value for the experimental group to match, this step reduces the selection bias. Among them, when conducting PSM, the samples of 114 resource-based cities in the past 15 years were divided into two categories for treating. One is the treated group i.e., “resource-based cities identified as ‘Broadband China’ demonstration cities between 2004 and 2018”, and the other category is the control group, that is, “resource-based cities that have not been identified as ‘Broadband China’ demonstration cities between 2004 and 2018”. When matching, the dependent variable of the model is whether the resource-based city is identified as a “Broadband China” demonstration city.

The matching and balance test was carried out using the city-level control variables as matching covariates, and the results are shown in Table 5. The results show that the t-statistic values of the covariates that affect resource-based cities becoming “Broadband China” demonstration cities and their carbon emission efficiency after matching are mostly insignificant, indicating that there is no significant difference between the treated group and the control group. At the same time, most of the standardized deviations of each variable after matching are less than 10%, which indicates that the covariates of the treated group and the control group have been basically balanced after matching, that is, the matching is effective. On the basis of satisfying the balance test, this paper further conducts PSM-DID regression, and the results are reported in columns (3) and (4) of Table 3. The results further show that there is no significant difference between the treated group and the control group after nuclear matching, and the “Broadband China” pilot policy has a significant effect on the carbon emission efficiency of resource-based cities in China, which further verifies the benchmark regression results.

#### 5.2.3. Placebo Test

In order to exclude the influence of other factors on the carbon emission efficiency of resource-based cities during the implementation of the policy, this paper draws on the practice of Cantoni et al. [82]. There are usually two types of placebo tests in DID, one is to change the point in time when the policy occurs, specifically including pre-treatment and post-treatment of the point in time when the policy occurs, when the placebo test serves the same purpose as the parallel trend test, both are to examine the significance of the time dummy variables and experimental group interaction term coefficients in the base regression before the policy occurs; the other is to randomize the experimental group, a certain number of times for the experimental group variables random sampling, and then observing whether the kernel density plots of the coefficients of the interaction terms or observations after randomization are concentrated around 0 and whether they significantly deviate from their true values. In addition, considering the length of the sample period can lead to conclusions drawn from random sampling that may not be true and affect the robustness of the conclusions. In this paper, the placebo test is conducted by random sampling based on the parallel trend through the following ideas: first, the sample data of the core explanatory variables are eliminated from the original data set; second, the data of the eliminated core explanatory variables are randomly disordered, and then the randomized core explanatory variables data are combined into the original data set that has been processed; third, the randomized core explanatory variables were put into the regression equation; fourth, the above operation steps were repeated 500 times; fifth, the coefficients and standard errors of the core explanatory variables were extracted separately from the 500 regression results, and finally the kernel density distribution of coefficients and t-values and the scatter plot of p-value coefficients were plotted.

Specifically, this paper randomly selects 36 cities from 114 resource-based cities as the treated group, and assumes that these 36 cities are identified as “Broadband China” demonstration cities, and other resource-based cities are the control group. A “pseudo-policy dummy” for a placebo test, and thus an erroneously estimated coefficient. Since the “pseudo-treat group” is randomly generated, the simulated policy dummy variable will not have an influence on the explained variable, and its erroneous estimated coefficient should be close to 0. This paper conducts 500 random samplings, and Figure 3 reports the significance and distribution of the estimated coefficients after 500 random samplings. It can be seen from Figure 3 that the distribution of the “pseudo-policy dummy variable” is concentrated around the zero point, and the corresponding P-value is greater than 0.1, which is in line with the expectations of the placebo test. This estimation result indirectly proves that the improvement of carbon emission efficiency of the treated group is indeed caused by the implementation of the “Broadband China” pilot policy, which further shows the robustness of the conclusions of this paper.

### 5.3. Heterogeneity Analysis

Although the influence of the “Broadband China” pilot policy on the carbon emission efficiency of resource-based cities has been demonstrated above, are there certain differences in the responses of different regions and different types of resource-based cities within the pilot scope to policy shocks? The discussion of this issue is conducive to an in-depth understanding of the conditions for the construction of information network infrastructure to promote the improvement of carbon emission efficiency in resource-based cities. In view of this, this paper further discusses and analyzes the heterogeneity of urban location and urban type. Furthermore, the model (5) is built for heterogeneity regression analysis:(5)$$CEE_{it}=\xi_{0}+\xi_{1}{\mathrm{broadband}}_{it}\times heter_{i}+\xi_{it}+\xi_{im}+\xi_{tm}+\sum_{j}\beta_{j}Control_{it}+\gamma_{t}+\eta_{i}+\varepsilon_{it}$$
where *heter* is the proxy variable of the dummy variable of location type (*area*) and the dummy variable of city type (*city*); *ξ*_1_ is the estimated coefficient of the interaction term between the dummy variable of “Broadband China” pilot policy, the dummy variable of location type and city type, indicating that the heterogeneous influence of the “Broadband China” pilot policy on its carbon emission efficiency due to the differences in the location and type of resource-based cities; *ξ_it_*, *ξ_im_* and *ξ_tm_* are the interaction effects, which are used to control the “Broadband China” pilot policy variables, the interaction terms of the city dummy variables and the heterogeneity dummy variables of the “Broadband China” pilot policy, and the interaction terms of the time dummy variables and the heterogeneity dummy variables of the “Broadband China” pilot policy; the other variables have the same meaning as in model (1).

#### 5.3.1. Heterogeneity Analysis of City Locations

Resource-based cities are important energy security bases in China and they play an irreplaceable role in economic development. However, the problem of unbalanced and insufficient development among resource-based cities in different locations is still prominent. Resource-based cities in the eastern region, supported by a series of experimental policies such as reform and opening up, took the lead in developing with convenient transportation and location advantages. Compared with resource-based cities in the central and western regions, factors such as economy, human resources, and scientific and educational resources are superior. In addition, the coastal resource-based cities have a good agricultural foundation, relatively developed industries, and more convenient water and land transportation, while the development of inland cities is relatively backward. Therefore, the differences in location advantages between resource-based cities in the east, central and western regions and the differences between resource-based cities in coastal and inland regions may lead to differences in the policy effects of “Broadband China” demonstration cities construction.

To this end, this paper constructed the location dummy variable (*area*) and assigned the value of cities in the eastern region to be 1 and cities in the central and western regions to be 0. Then, this variable was multiplied by the dual difference item (*broadband*) of the pilot policy into the regression. The results are shown in column (1) and column (2) of Table 6. It can be seen from Table 6 that in column (1), the regression coefficient of the multiplication term of the “Broadband China” pilot policy and the location dummy variable in column (1) are significantly positive at the 1% level, which indicates that the “Broadband China” pilot policy has a significant positive influence on the carbon emission efficiency of resource-based cities in eastern China. The main reason is that the resource-based cities in eastern China are more economically developed and have more abundant human resources and innovation resources. The “Broadband China” pilot policy is more likely to play a role in promoting the development of carbon emission technology. Additionally, economic development of the central and western regions and the inland region of resource-based cities is relatively backward, and science and education resources and innovation capability are relatively weak. Although the “Broadband China” pilot policy has accelerated the network technology spillover and spatial diffusion of innovative elements and information knowledge to a certain extent, limited by economic scale, resource endowment, and geographical location constraints, the endogenous driving force of innovative technologies for decarbonization and emission reduction is still insufficient, which makes it difficult to effectively exert the agglomeration driving effect of broadband infrastructure construction, which in turn has an influence on the policy effect of “Broadband China” pilot policy.

#### 5.3.2. Heterogeneity Analysis of City Types

According to the “National Sustainable Development Plan for Resource-Based Cities (2013–2020)” (referred to as the “Plan”) issued by the State Council in 2013, resource-based cities are classified into four types—growth, mature, declining, and regenerative—according to their resource security capabilities and economic and socially sustainable development capabilities. This paper further divides resource-based cities into three categories—growth period, mature period, and decline period—according to their life cycle characteristics. The resource-based cities in the growth stage are the growth and regeneration resource-based cities listed in the plan, and the resource-based cities in the mature and declining stages are the mature and declining resource-based cities listed in the plan, respectively. Additionally, based on model (5), regression is carried out to analyze the heterogeneity of the influence of the “Broadband China” pilot policy on the carbon emission efficiency of different types of resource-based cities. The regression results are shown in Table 7.

From Table 7, it can be seen that for declining resource-based cities, the “Broadband China” pilot policy has a significant positive effect on its carbon emission efficiency, and the regression coefficient is significant at the level of 1%, while for the growth and maturity stages, resource-based cities have no significant influence. The main reasons are: First, declining resource-based cities are faced with depletion of resources, serious environmental damage, and great pressure on their transformation and development. The implementation of the “Broadband China” strategy is to use network information technology for transformation and development. In addition, the foundation of innovation and development in the early stage is relatively solid, and the “Broadband China” pilot policy will play a more significant role in promoting its green and low-carbon transformation and improving carbon emission efficiency. Second, for growth-type and renewable resource-based cities in the growth stage, their development is in the growth stage, the urban infrastructure construction and talent team cultivation need to be improved, and the overall innovation capability foundation is weak, so the promotion effect brought by the “Broadband China” pilot policy is not significant. Third, for mature resource-based cities in the mature stage, their economic and social development level is relatively high, and the network information facilities are also more complete. The implementation of the “Broadband China” pilot policy is more “icing on the cake”, so the “Broadband China” pilot policy does not have a significant effect on it.

## 6. Mechanism Analysis

Table 8 shows the regression results based on the intermediate effect regression model (2) and (3). It can be seen from Table 8 that the “Broadband China” pilot policy has significantly promoted the upgrading of the industrial structure of resource-based cities, the accumulation of human capital, and the improvement of the level of urban innovation. Further from the regression results of column (3), column (5), and column (7) based on model (3), the coefficient value of “Broadband China” pilot policy has declined, as have the industrial structure upgrade, human capital and innovation level. The regression coefficients are significantly positive at the levels of 10%, 5%, and 1%, respectively, that is, the industrial structure upgrade, human capital and innovation level play a partial intermediary effect in the promotion of the carbon emission efficiency of resource-based cities by the “Broadband China” pilot policy. After further calculation, it can be concluded that the proportion of the mediation effect to the total effect under the three paths of industrial structure upgrade, human capital, and innovation level is 9.64%, 14.51%, and 10.42%, respectively. The above conclusions show that the construction of information network infrastructure with broadband as the core provides the material basis and technical support for the transmission and transfer of information data elements, creates favorable conditions for the development of new low-carbon technologies, and is conducive to promoting the transforming and upgrading of the industrial structure of resource-based cities, attracting more talents, stimulating urban innovation vitality, and promoting improvement in the carbon emission efficiency of resource-based cities.

## 7. Conclusions and Discussion

### 7.1. Conclusions

This paper takes the “Broadband China” pilot policy as a quasi-natural experiment and uses the DID method to investigate the influence of broadband infrastructure on the carbon emission efficiency of resource-based cities and the influence mechanism. The main conclusions are as follows: First, the broadband infrastructure has a significant role in promoting the carbon emission efficiency of resource-based cities. Secondly, the influence of broadband infrastructure on the carbon emission efficiency of resource-based cities has city location heterogeneity and city type heterogeneity. Finally, the “Broadband China” pilot policy can promote the improvement of carbon emission efficiency of resource-based cities by promoting the upgrading of industrial structure, the accumulation of human capital, and the improvement of urban innovation level.

Combining the above conclusions, the policy implications are as follows: **First**, local governments should actively promote the construction of broadband infrastructure. All localities should seize the opportunity of the “Broadband China” pilot policy to increase investment in broadband network infrastructure development. At the same time, it is necessary, in order to achieve high-quality development of city, to vigorously strengthen the construction of 5G networks, data centers, and the Internet of Things, give full range to the functions of broadband infrastructure, and activate the organic connection between urban innovation subjects, elements, and industries, truly internalize the new generation of information technology as the driving force for urban innovation and low-carbon development. **Second**, policies should be classified according to different locations and types of resource-based cities. Combined with various factors such as the development level of network infrastructure, technology level, and geographical location in each region, the “Broadband China” strategic action is carried out in a targeted manner to better play the role of broadband infrastructure construction in improving the carbon emission efficiency of resource-based cities. On the one hand, for resource-based cities in different locations, the government should steadily advance the pace of broadband infrastructure in the eastern region and increase the financial and technological support needed for the development of the “Broadband” infrastructure in the central and western regions. On the other hand, for resource-based cities of different city types, the government should focus on supporting resource-based cities in the recession period and guide them to vigorously develop the information technology industry, so as to stimulate their latecomer advantages. Meanwhile, it is necessary for resource-based cities in the growth stage to accelerate the cultivation of innovation talent, and gradually improve the construction of broadband infrastructure. Additionally, for resource-based cities in the mature stage, the steady promotion of the construction of broadband infrastructure should be continued. **Third**, we should make full use of the influence mechanism of broadband infrastructure construction on the carbon emission efficiency of resource-based cities. Above all, according to the development strategies of different cities, more resource-based cities in the dilemma of industrial transformation should be included in the pilot scope, so that they can rely on the construction of network infrastructure to stimulate the vitality of industrial transformation, and realize the transformation of manufacturing from the traditional raw material manufacturing industry with high pollution and high energy consumption to the high-end equipment manufacturing industry with high intelligent processing degree. Additionally, this will further promote the transformation of the service industry to the high-end industry of the value chain, form a multi-dimensional transformation and upgrading path of the industrial structure, and help reduce city pollution. Furthermore, relevant departments should strengthen the introduction of talents, guiding the inflow of high-quality human capital into high-tech industries on the basis of improving the introduction mechanism of high-tech talents. At the same time, relevant departments also need to pay attention to the secondary training of talents, improve the training and reward mechanism of talents, and give full play to the mechanism of capital accumulation on carbon emission efficiency improvement. Lastly, local governments should actively guide and encourage scientific research institutions and enterprises to use broadband networks for informatization, digitization, and intelligent construction, in particular, increasing support for R&D institutions, enterprises, and scientific research institutes related to cleaner production, pollution prevention and green ecology, and building a comprehensive network innovation research and development platform. Promoting the formation of the green innovation research and development platform is especially necessary to fully stimulate the vitality of city green innovation, improve the city green technology innovation ability, and realize the improvement of carbon emission efficiency of resource-based cities.

### 7.2. Discussion

There is still much room for improvement in this study. In this paper, a quasi-natural experiment is constructed through the “Broadband China” pilot policy to assess the effects of broadband infrastructure development on the carbon efficiency of resource-based cities and the mechanisms of effects. Although the findings of this paper are supported by a rigorous analytical approach, more in-depth exploration is needed to determine whether the policy choice can truly cover all aspects of broadband development and whether there are more appropriate policies. In addition, the research object of this paper is limited to resource-based cities with special development characteristics, and different countries, regions and city clusters are at different levels of development, and further research is needed to see whether the findings of this paper can be extended to countries, other types of cities or regions. As the world enters the era of digital economy, the development of digital economy cannot be separated from the development of broadband network, and further evaluation of the effects of broadband infrastructure and other policies related to digital economy is of great significance to the national digital economy and digital transformation. For example, building high-tech smart cities has become one of the important ways to solve urban problems such as environmental pollution, and the effects of smart city construction on carbon emission efficiency can be further analyzed.

## Figures and Tables

**Figure 1 ijerph-19-06734-f001:**
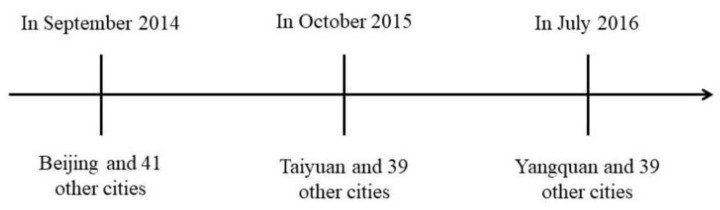
“Broadband China” city development process.

**Figure 2 ijerph-19-06734-f002:**
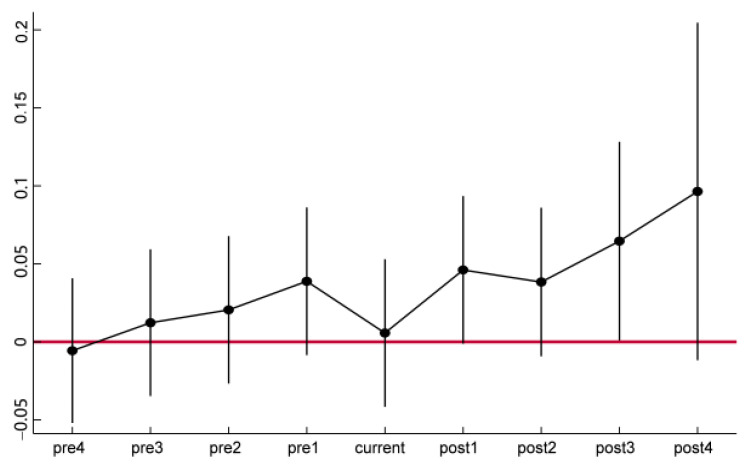
Parallel trend test chart.

**Figure 3 ijerph-19-06734-f003:**
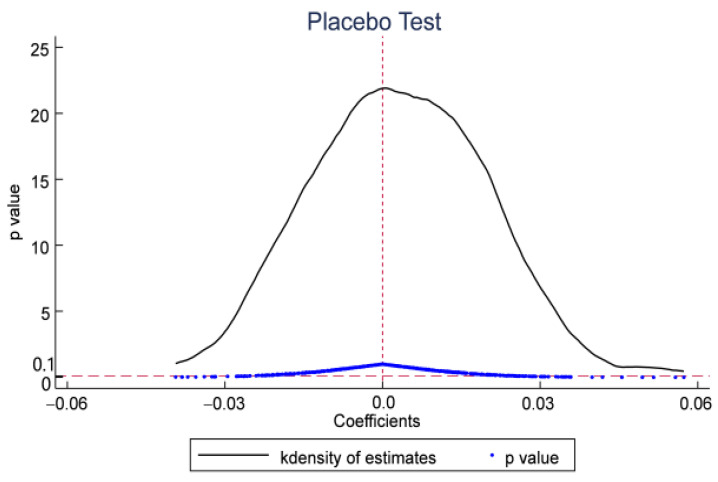
Placebo test.

**Table 1 ijerph-19-06734-t001:** Input–output index system of carbon emission efficiency.

Index	Variable	Unit	Calculation Method
input	fixed capital stock	million	$K_{it}=K_{it-1}(1-\delta_{it})+I_{it}$ ^1^
	employees	10,000 people	-
	social electricity consumption	10,000 kWh	-
desired output	GDP	million	-
undesired output	CO_2_ emission	10,000 t	$Q=C_{1}+C_{2}+C_{3}=kE_{1}+vE_{2}+\varphi (\eta E_{3})$ ^2^

^1^ *K_it_* is fixed capital stock of the current year; *I_it_* is fixed asset investment in the current year; *δ_it_* is the rate of economic depreciation, which is 9.6%. ^2^
*Q* represents total carbon dioxide emissions; *C*_1_, *C*_2_ and *C*_3_ are carbon dioxide emissions caused by LPG, gas and electricity consumption of the whole society, respectively; *K* is the CO_2_ conversion coefficient of LPG, *E*_1_ is the consumption of LPG; *V* is the CO_2_ conversion coefficient of gas, and *E*_2_ is gas consumption; *E*_3_ is annual electricity consumption; *η* is the proportion of coal power in the total power generation; *φ* is the GHG emission coefficient of coal power fuel chain.

**Table 2 ijerph-19-06734-t002:** Descriptive statistics of variables.

Variable Category	Variable Name	Variable Unit	Observation	Mean	Mean Standard	Minimum	Maximum
Explained variable	*CEE_it_*	-	1710	0.463	0.221	0.157	1.118
Core explanatory variable	*broadband*	-	1710	0.0778	0.268	0	1
Intermediary variables	*industrs*	%	1710	35.07	7.640	8.580	85.34
	*humancs*	%	1710	0.00879	0.00779	0	0.0576
	*innova*	%	1710	0.107	0.0731	0	0.667
Control variables	*traff*	million	1710	0.941	0.812	0.0500	6.090
	*unemprate*	%	1710	3.502	2.060	0.272	22.23
	*perra*	m^2^	1710	14.11	7.334	2.250	60.07
	*cityscale*	%	1710	0.334	0.220	0.0611	1
	*education*	%	1710	18.26	4.317	1.69 × 10^−5^	37.74
	*iGDP*	%	1710	11.06	4.994	−19.38	37.69
	*phycl*	%	1710	2.489	4.933	0	63.95

**Table 3 ijerph-19-06734-t003:** Benchmark regression results.

Variable	*CEE_it_*			
	(1)	(2)	(3)	(4)
*broadband*	0.0280 *	0.0290 **	0.0245 *	0.0259 *
	(0.0147)	(0.0148)	(0.0148)	(0.0149)
*traff*		−0.0388 ***		−0.0391 ***
		(0.0091)		(0.0091)
*unemprate*		0.0074 ***		0.0049 **
		(0.0020)		(0.0023)
*perra*		−0.0020 **		−0.0021 **
		(0.0009)		(0.0009)
*cityscale*		−0.0125		−0.0110
		(0.0541)		(0.0541)
*education*		0.0026 **		0.0027 **
		(0.0012)		(0.0013)
*iGDP*		−0.0016 *		−0.0019 *
		(0.0010)		(0.0010)
*phycl*		−0.0020 **		−0.0020 **
		(0.0008)		(0.0008)
*_cons*	0.4024 ***	0.4078 ***	0.4053 ***	0.4255 ***
	(0.0117)	(0.0389)	(0.0118)	(0.0396)
*N*	1710	1710	1691	1691

Note: Robust standard errors are reported in parentheses; *, **, and *** are significant at the 10%, 5%, and 1% levels, respectively.

**Table 4 ijerph-19-06734-t004:** Parallel trend test regression results.

Period	*CEE_it_*	Period	*CEE_it_*
pre4	−0.0056	post1	0.0461 *
	(−0.2379)		(1.9100)
pre3	0.0123	post2	0.0384
	(0.5125)		(1.5810)
pre2	0.0205	post3	0.0646 **
	(0.8515)		(1.9921)
pre1	0.0389	post4	0.0965 *
	(1.6103)		(1.7484)
current	0.0057	_cons	0.4024 ***
	(0.2354)		(34.5181)
*N*	1710	*N*	1710

Note: Value T statistics are reported in parentheses; *, **, and *** are significant at the 10%, 5%, and 1% levels, respectively.

**Table 5 ijerph-19-06734-t005:** PSM and balance test results.

Variable	Coefficient	Sample	Mean		% Bias	t	*p* > │t│
			Treated	Control			
*traff*	0.2076	Unmatched	0.9797	0.9239	7.3	1.32	0.18
	(0.0695)	Matched	0.9897	1.0592	−9.0	−1.21	0.22
*unemprate*	−0.1008	Unmatched	3.3890	3.5544	−8.1	−1.54	0.12
	(0.0316)	Matched	3.3934	3.3930	0.0	0.00	0.99
*perra*	−0.0347	Unmatched	15.7280	13.3560	31.7	6.29	0.00
	(0.0078)	Matched	15.6740	15.6750	−0.0	−0.00	1.00
*cityscale*	2.6947	Unmatched	0.4251	0.2917	60.2	12.17	0.00
	(0.2703)	Matched	0.4165	0.4143	1.0	0.14	0.88
*education*	−0.0697	Unmatched	17.1780	18.7540	−36.0	−7.12	0.00
	(0.0138)	Matched	17.2720	17.7460	−10.8	−1.76	0.07
*iGDP*	0.0485	Unmatched	11.3710	10.9180	9.0	1.75	0.08
	(0.0117)	Matched	11.2760	11.3610	−1.7	−0.28	0.78
*phycl*	0.0183	Unmatched	3.1807	2.1700	20.0	3.95	0.00
	(0.0113)	Matched	3.2247	3.3158	−1.8	−0.24	0.81

Note: Robust standard errors are reported in parentheses.

**Table 6 ijerph-19-06734-t006:** Heterogeneity regression results for different city locations.

Variables	*CEE_it_*			
	(1)	(2)	(4)	(5)
*broadband* × *east*	0.0680 ***			
	(0.0263)			
*broadband* × *mwest*		0.0091		
		(0.0162)		
*broadband* × *coastal*			0.0680 ***	
			(0.0263)	
*broadband* × *inland*				0.0091
				(0.0162)
*traff*	−0.0389 ***	−0.0371 ***	−0.0389 ***	−0.0371 ***
	(0.0090)	(0.0091)	(0.0090)	(0.0091)
*unemprate*	0.0075 ***	0.0075 ***	0.0075 ***	0.0075 ***
	(0.0020)	(0.0020)	(0.0020)	(0.0020)
*perra*	−0.0021 **	−0.0021 **	−0.0021 **	−0.0021 **
	(0.0009)	(0.0009)	(0.0009)	(0.0009)
*cityscale*	−0.0198	−0.0060	−0.0198	−0.0060
	(0.0542)	(0.0541)	(0.0542)	(0.0541)
*education*	0.0027 **	0.0028 **	0.0027 **	0.0028 **
	(0.0012)	(0.0012)	(0.0012)	(0.0012)
*iGDP*	−0.0014	−0.0017 *	−0.0014	−0.0017 *
	(0.0010)	(0.0010)	(0.0010)	(0.0010)
*phycl*	−0.0018 **	−0.0020 **	−0.0018 **	−0.0020 **
	(0.0008)	(0.0009)	(0.0008)	(0.0009)
*_cons*	0.4061 ***	0.4010 ***	0.4061 ***	0.4010 ***
	(0.0387)	(0.0388)	(0.0387)	(0.0388)
*N*	1710	1710	1710	1710

Note: Robust standard errors are reported in parentheses; *, **, and *** are significant at the 10%, 5%, and 1% levels, respectively.

**Table 7 ijerph-19-06734-t007:** Heterogeneity regression results for different city types.

Variables	*CEE_it_*		
	(1)	(2)	(3)
*broadband* × *growth*	−0.0097		
	(0.0226)		
*broadband* × *mature*		0.0164	
		(0.0200)	
*broadband* × *declining*			0.0846 ***
			(0.0275)
*traff*	−0.0364 ***	−0.0372 ***	−0.0369 ***
	(0.0091)	(0.0090)	(0.0090)
*unemprate*	0.0076 ***	0.0075 ***	0.0078 ***
	(0.0020)	(0.0020)	(0.0020)
*perra*	−0.0021 **	−0.0020 **	−0.0021 **
	(0.0009)	(0.0009)	(0.0009)
*cityscale*	−0.0053	−0.0091	−0.0038
	(0.0541)	(0.0542)	(0.0539)
*education*	0.0029 **	0.0028 **	0.0026 **
	(0.0012)	(0.0012)	(0.0012)
*iGDP*	−0.0017 *	−0.0017 *	−0.0016 *
	(0.0010)	(0.0010)	(0.0010)
*phycl*	−0.0019 **	−0.0019 **	−0.0018 **
	(0.0008)	(0.0008)	(0.0008)
*_cons*	0.3979 ***	0.4013 ***	0.4018 ***
	(0.0388)	(0.0388)	(0.0386)
*N*	1710	1710	1710

Note: Robust standard errors are reported in parentheses; *, **, and *** are significant at the 10%, 5%, and 1% levels, respectively.

**Table 8 ijerph-19-06734-t008:** Mechanism regression results.

Variables	*CEE_it_*	*Industrs*	*CEE_it_*	*Humancs*	*CEE_it_*	*Innova*	*CEE_it_*
	(1)	(2)	(3)	(4)	(5)	(6)	(7)
*broadband*	0.0290 **	1.6443 ***	0.0262 *	0.0019 ***	0.0247 *	0.0202 ***	0.0260 *
	(0.0148)	(0.4194)	(0.0148)	(0.0004)	(0.0149)	(0.0073)	(0.0148)
*industrs*			0.0017 *				
			(0.0009)				
*humancs*					2.2140 **		
					(1.0125)		
*innova*							0.1496 ***
							(0.0510)
*traff*	−0.0388 ***	−0.3918	−0.0381 ***	−0.0001	−0.0386 ***	−0.0003	−0.0387 ***
	(0.0091)	(0.2579)	(0.0091)	(0.0002)	(0.0091)	(0.0045)	(0.0091)
*unemprate*	0.0074 ***	0.2281 ***	0.0070 ***	0.0001 *	0.0072 ***	−0.0011	0.0076 ***
	(0.0020)	(0.0571)	(0.0020)	(0.0000)	(0.0020)	(0.0010)	(0.0020)
*perra*	−0.0020 **	−0.0019	−0.0020 **	0.0000	−0.0021 **	0.0003	−0.0021 **
	(0.0009)	(0.0254)	(0.0009)	(0.0000)	(0.0009)	(0.0004)	(0.0009)
*cityscale*	−0.0125	−2.6749 *	−0.0079	0.0018	−0.0164	−0.0041	−0.0119
	(0.0541)	(1.5369)	(0.0541)	(0.0013)	(0.0541)	(0.0267)	(0.0540)
*education*	0.0026 **	0.1124 ***	0.0024 *	0.0001 *	0.0025 **	0.0022 ***	0.0023 *
	(0.0012)	(0.0352)	(0.0012)	(0.0000)	(0.0012)	(0.0006)	(0.0012)
*iGDP*	−0.0016 *	−0.2066 ***	−0.0012	0.0000	−0.0017 *	0.0006	−0.0017 *
	(0.0010)	(0.0272)	(0.0010)	(0.0000)	(0.0010)	(0.0005)	(0.0010)
*phycl*	−0.0020 **	0.0276	−0.0020 **	−0.0000	−0.0020 **	0.0014 ***	−0.0022 ***
	(0.0008)	(0.0241)	(0.0008)	(0.0000)	(0.0008)	(0.0004)	(0.0008)
*_cons*	0.4078 ***	33.2560 ***	0.3506 ***	0.0024 **	0.4024 ***	0.0657 ***	0.3980 ***
	(0.0389)	(1.1044)	(0.0488)	(0.0010)	(0.0389)	(0.0192)	(0.0389)
*N*	1710	1710	1710	1710	1710	1710	1710

Note: Robust standard errors are reported in parentheses; *, **, and *** are significant at the 10%, 5%, and 1% levels, respectively.

## Data Availability

The data presented in this study can be collected and found in publicly available databases, and the data sources are mainly the “China City Statistical Yearbook”, the “Urban Construction Statistical Yearbook”, and the Statistical Bulletins of National Economic and Social Development of prefecture-level cities, as follows:
The relevant data sources involved in the carbon emission efficiency (*CEE_it_*) in this paper(1) Fixed capital stock: the data used for the amount of fixed asset investment are from the “China City Statistical Yearbook” from 2005 to 2017 and the Statistical Bulletin of National Economic and Social Development of resource-based cities at all levels in China from 2017–2018, and the economic depreciation rate involved in inventorying fixed assets (9.6%) is from a literature study by Zhang Jun et al. at the link [https://kns.cnki.net/kcms/detail/detail.aspx?FileName=JJJYJ200410004&DbName=CJFQ2004].(2) Number of employees: from “China City Statistical Yearbook” for resource-based cities at all levels in China from 2005 to 2019.(3) Social electricity consumption: from “China City Statistical Yearbook” for resource-based cities at all levels in China from 2005 to 2019.(4) GDP: from “China City Statistical Yearbook” for resource-based cities at all levels in China from 2005 to 2019.(5) CO_2_ emissions: the social electricity consumption, total natural gas and LPG consumption involved in the calculation of CO_2_ emissions are from the “China City Statistical Yearbook” for resource-based cities at all levels in China from 2005 to 2019; the CO_2_ emission factors for social electricity consumption, total natural gas and LPG are from the literature study by Ren et al. at the link [https:// kns.cnki.net/kcms/detail/detail.aspx?FileName=ZGRZ202004011&DbName=CJFQ2020]Data sources for the core explanatory variables in this paperThe dummy variables of the "Broadband China" pilot policy are set based on the list of 3 batches of “Broadband China” demonstration cities (city groups) released by the Ministry of Industry and Information Technology of the People’s Republic of China in 2015 and 2016, and the links to the websites are [https://www.miit.gov.cn/ztzl/lszt/qltjkdzg/yw/art/2014/art_032c5267911a4bfdb06fd206da5863f0.html; https://www.miit.gov.cn/ztzl/lszt/qltjkdzg/yw/art/2015/art_71703efdf3ec4fedb7fae0c924e9606b.html; https://www.miit.gov.cn/jgsj/txs/wlfz/art/2020/art_f9f5db18c95a48a498e487a74699312c.html]Data sources for the mechanism variables in this paper(1) Tertiary industry output as a share of GDP (*industrs*): both tertiary industry output and GDP data were obtained from the “China City Statistical Yearbook” for resource-based cities at all levels in China from 2005 to 2019.(2) Number of students enrolled in higher education as a proportion of total population (*humancs*): the source of the number of students enrolled in higher education is the number of general undergraduate and college students enrolled in the “China City Statistical Yearbook” from 2005 to 2019 for resource-based cities at all levels in China.(3) The proportion of city invention patents to total patents (*innova*): Both city invention patents and total patents data are obtained from the patent search system of the China National Intellectual Property Administration of the People’s Republic of China, URL link [http://www.cnipa.gov.cn/]Data sources for the control variables in this paper(1) The number of employees in the transportation, storage and postal industry (*traff*): from “China City Statistical Yearbook” for resource-based cities at all levels in China from 2005 to 2019.(2) Urban registered unemployment rate (*unemprate*): from “China City Statistical Yearbook” for resource-based cities at all levels in China from 2005 to 2019.(3) The urban road area per capita (*perra*): from the China “Urban Construction Statistical Yearbook”, 2004 to 2018.(4) The year-end city municipal area population share (*cityscale*): population of urban municipal districts at the end of the year is from “China City Statistical Yearbook” for resource-based cities at all levels in China from 2005 to 2019.(5) The proportion of education expenditure to total fiscal expenditure (*education*): Education expenditures and total fiscal expenditures are from the “China City Statistical Yearbook” for resource-based cities at all levels in China from 2005 to 2019.(6) GDP growth rate (*iGDP*): from “China City Statistical Yearbook” for resource-based cities at all levels in China from 2005 to 2019.(7) The proportion of fixed asset investment to GDP (*phycl*): The values of fixed asset investment and GDP are from the “China City Statistical Yearbook” for resource-based cities at all levels in China from 2005 to 2019. The relevant data sources involved in the carbon emission efficiency (*CEE_it_*) in this paper (1) Fixed capital stock: the data used for the amount of fixed asset investment are from the “China City Statistical Yearbook” from 2005 to 2017 and the Statistical Bulletin of National Economic and Social Development of resource-based cities at all levels in China from 2017–2018, and the economic depreciation rate involved in inventorying fixed assets (9.6%) is from a literature study by Zhang Jun et al. at the link [https://kns.cnki.net/kcms/detail/detail.aspx?FileName=JJJYJ200410004&DbName=CJFQ2004]. (2) Number of employees: from “China City Statistical Yearbook” for resource-based cities at all levels in China from 2005 to 2019. (3) Social electricity consumption: from “China City Statistical Yearbook” for resource-based cities at all levels in China from 2005 to 2019. (4) GDP: from “China City Statistical Yearbook” for resource-based cities at all levels in China from 2005 to 2019. (5) CO_2_ emissions: the social electricity consumption, total natural gas and LPG consumption involved in the calculation of CO_2_ emissions are from the “China City Statistical Yearbook” for resource-based cities at all levels in China from 2005 to 2019; the CO_2_ emission factors for social electricity consumption, total natural gas and LPG are from the literature study by Ren et al. at the link [https:// kns.cnki.net/kcms/detail/detail.aspx?FileName=ZGRZ202004011&DbName=CJFQ2020] Data sources for the core explanatory variables in this paper The dummy variables of the "Broadband China" pilot policy are set based on the list of 3 batches of “Broadband China” demonstration cities (city groups) released by the Ministry of Industry and Information Technology of the People’s Republic of China in 2015 and 2016, and the links to the websites are [https://www.miit.gov.cn/ztzl/lszt/qltjkdzg/yw/art/2014/art_032c5267911a4bfdb06fd206da5863f0.html; https://www.miit.gov.cn/ztzl/lszt/qltjkdzg/yw/art/2015/art_71703efdf3ec4fedb7fae0c924e9606b.html; https://www.miit.gov.cn/jgsj/txs/wlfz/art/2020/art_f9f5db18c95a48a498e487a74699312c.html] Data sources for the mechanism variables in this paper (1) Tertiary industry output as a share of GDP (*industrs*): both tertiary industry output and GDP data were obtained from the “China City Statistical Yearbook” for resource-based cities at all levels in China from 2005 to 2019. (2) Number of students enrolled in higher education as a proportion of total population (*humancs*): the source of the number of students enrolled in higher education is the number of general undergraduate and college students enrolled in the “China City Statistical Yearbook” from 2005 to 2019 for resource-based cities at all levels in China. (3) The proportion of city invention patents to total patents (*innova*): Both city invention patents and total patents data are obtained from the patent search system of the China National Intellectual Property Administration of the People’s Republic of China, URL link [http://www.cnipa.gov.cn/] Data sources for the control variables in this paper (1) The number of employees in the transportation, storage and postal industry (*traff*): from “China City Statistical Yearbook” for resource-based cities at all levels in China from 2005 to 2019. (2) Urban registered unemployment rate (*unemprate*): from “China City Statistical Yearbook” for resource-based cities at all levels in China from 2005 to 2019. (3) The urban road area per capita (*perra*): from the China “Urban Construction Statistical Yearbook”, 2004 to 2018. (4) The year-end city municipal area population share (*cityscale*): population of urban municipal districts at the end of the year is from “China City Statistical Yearbook” for resource-based cities at all levels in China from 2005 to 2019. (5) The proportion of education expenditure to total fiscal expenditure (*education*): Education expenditures and total fiscal expenditures are from the “China City Statistical Yearbook” for resource-based cities at all levels in China from 2005 to 2019. (6) GDP growth rate (*iGDP*): from “China City Statistical Yearbook” for resource-based cities at all levels in China from 2005 to 2019. (7) The proportion of fixed asset investment to GDP (*phycl*): The values of fixed asset investment and GDP are from the “China City Statistical Yearbook” for resource-based cities at all levels in China from 2005 to 2019.

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
