# Peer review of "The Effects of Broadband Infrastructure on Carbon Emission Efficiency of Resource-Based Cities in China: A Quasi-Natural Experiment from the “Broadband China” Pilot Policy"

_ijerph, 2022, doi:10.3390/ijerph19116734_

Round 1

Reviewer 1 Report

The authors want to demonstrate in the article that the broadband infrastructure has an effect on carbon emission efficiency.

The results are based on a formula to calculate CO2 emissions (Table 1). How solid or reliable is this formula? Are there any other ways to calculate CO2 emission is in a city?

It is unclear how the values for broadband infrastructure were derived. Is it fixed and mobile broadband? Which type of mobile broadband infrastructure? 4G, 5G? Which type of fixed network infrastructure? FTTH? Copper based?

It is not clear how the broadband infrastructure in a given city was calculated.

Becase the two main input parameters are not clear enough, i.e. broadband infrastructure and CO2 emissions calculations, the recommendation is that the article is rejected. Perhaps the authors could explain these input parameters much better in a potential future new version of the article, which could also be submitted to this journal. Also the limitations of the approach taken to derive both values should be explained in the article.

Reviewer 2 Report

This study uses the difference-in-differences (DID) method to identify the effects of the “Broadband” infrastructure on the carbon emission efficiency of resource-based cities in China.  This is a very interesting topic, but, in order to improve the quality of this research, I think there are still areas for improvement.

  1. In introduction section, it is suggested that the literature review and the introduction should be written separately so as to expand the citation of the current literature.
  2. In the research hypothesis section, why does broadband infrastructure impact carbon efficiency in three ways, rather than more or less? What is this based on? It is necessary to elaborate.
  3. Line 243, SBM is one of the DEA models, and the two are not in a parallel relationship. In addition, it is recommended to add a detailed introduction to the SBM model settings in the appendix.
  4. In Table 1, what is the depreciation rate of fixed capital stock and on what basis, there should be a clear statement.
  5. The results are not discussed in the context of previous findings due to the lack of a proper current literature review.
  6. It is suggested to move the intermediate effect model to section 3.1 as a part of the research design to ensure the integrity of the research design, rather than scattered in different parts of the manuscript.
  7. It is suggested to add on limitations and future research suggestions.

Reviewer 3 Report

I would see a problem already with this statement from the Abstract: “The results show that the “Broadband” policy has a significant effect on the improvement of carbon emission efficiency of resource-based cities”. But the purpose of this paper is to “identify the effects of the “Broadband” infrastructure on the carbon emission efficiency of resource-based cities.”. Even if you think of a “Broadband” policy as belonging to “Broadband” infrastructure, I do not think you a “Broadband” policy can be used to generalize “Broadband” infrastructure. I would suggest you reconsider the purpose of this paper, since you only take “Broadband” policy into consideration.

The motivation should be strengthened significantly. The link between resource-based cities and broadband infrastructure is not clearly expressed in the introduction section. Does broadband infrastructure produce higher carbon emission in resource-based cities than in ordinary cities?

There is no explanation for E3 on page 6. If my understanding is correct, E3 is coal power consumption, why only here you add a proportion for this part (coal power consumption). You should explain how the calculation works here.

Again, I cannot tell the connection between control variables and Broadband” infrastructure. You should also give reasonable explanations for the controls variables. For example, can “the number of people employed in transportation, storage, and postal service (traff)” really be used as a representative index to measure the level of transportation?

Reviewer 4 Report

Methodology: Sampling needs to be elaborated further and clearly. How are these samplings selected? The process of sample selection should be clearly elaborated. Not just mention the total sample.

Include a discussion section in your manuscript.

In the conclusion section, the limitations of the research need to be discussed.

Round 2

Reviewer 1 Report

The authors did not manage to explain in detail in the article how the value for broadband was derived: What is in detail the broadband infrastructure? Fixed, mobile broadband infrastructure? They refer broadly to the “Broadband China pilot policy”? What is this pilot policy in detail? How many Central Offices, Mobile Stations, etc ware considered for the different cities. Is 5G considered? Fiber to the Home (FTTH)? Or the current cupper-based infrastructure for fixed networks?

As the is the most important input parameter for the study, the values of the broadband infrastructure, was not clearly explained in the reviewed version of the article, then the whole contribution from a scientific point of view is not clear. The article should unfortunately be rejected according to the reviewer.

Author Response

Thank you very much for the reviewer’s comments! Your comments are of great value in highlighting our research contributions, we combine your suggestions and current academic research related to broadband and find that the definition of broadband infrastructure is not clear, it is a dynamic and evolving concept, which includes broadband network infrastructure involved in the adoption of wired access such as xDSL, FTTx and wireless access such as 3G, LTE, broadband satellite, etc. In the introduction of the revised manuscript (Section1), we explain the broadband infrastructure, and in the literature review (Section2), we add relevant studies on broadband to further clarify the value of broadband infrastructure. In addition, we introduce the “Broadband China” pilot policy in more detail in the revised draft (Section3.1), mainly including the policy background, the milestones of the policy implementation and the number of cities, but due to the availability of data, there are some limitations in this study, and we will further clarify the different In the next study, we will further clarify the range of the number of Central Offices, Mobile Stations, etc. in different cities.( See revised manuscript marked yellow section)